# Influencing Factors of Psychosocial Stress among Korean Adults during the COVID-19 Outbreak

**DOI:** 10.3390/ijerph18116153

**Published:** 2021-06-07

**Authors:** Jina Kim, Ogcheol Lee

**Affiliations:** Red Cross College of Nursing, Chung-Ang University, Seoul 06974, Korea; jinaword@cau.ac.kr

**Keywords:** COVID-19, knowledge, health belief, psychosocial stress, resilience

## Abstract

The restriction of an individual’s daily life due to the strengthening of quarantine and lockdown increases psychosocial stress. This study aimed to determine the factors that influence psychosocial stress during a period of strict quarantine and lockdown to curb the COVID-19 pandemic in Korea. A total of 338 adults participated in a cross-sectional online survey conducted from 19–25 May 2020, which measured knowledge of COVID-19, health belief, resilience, and psychosocial stress. According to the results, there was no difference between the participants’ scores from the Daegu area (with concentrated confirmed COVID-19 cases) and the non-Daegu area except for health belief. Eighty-two percent of participants constituted the high-risk group for psychosocial stress. Individual resilience was positively correlated with health belief and negatively correlated with psychosocial stress (*p* < 0.001). Further, the following factors affected the level of psychosocial stress: resilience, subjective health status, and monthly household income, with an explanatory power of 39.8%. Therefore, those with higher subjective health and higher monthly household income experienced higher psychosocial stress, whereas higher resilience indicated lower psychosocial stress.

## 1. Introduction

Coronavirus disease-2019 (COVID-19) originated in Wuhan, China, in December 2019. On 31 January 2020, the World Health Organization (WHO) declared COVID-19 a pandemic [1]. At present, we are witnessing the third wave of outbreaks caused by variants of the virus. Quarantine and lockdown are common strategies utilized globally to tackle the spread of this infection; however, they have produced undesirable social issues [2,3,4,5].

On 20 January 2020, the first confirmed case of COVID-19 was observed among the returnees to Korea who visited Wuhan, China. Thereafter, on 19 February, confirmed cases were reported from a religious group in the Daegu area (Gyeongbuk Province surrounding Daegu) and consequently on 23 February, the alert level in the region was raised to “highest” (level 4) [6]. As of 29 February 2020, the number of confirmed cases in the Daegu area accounted for 91% of all the confirmed cases in Korea (Korea Center for Disease Control and Prevention, KCDC [7]; Figure 1).

Measures against COVID-19 have altered everyday life, increased the fear of infection, and negatively influenced psychological well-being [2,3,8]. These measures have especially affected patients, self-quarantined individuals, and healthcare providers [9,10,11]. Negative psychological impacts may appear in the form of anxiety, frustration, boredom, and loneliness. Some of the influencing factors in adapting and overcoming these changes are an individual’s resilience [3,8,12], COVID-19-related knowledge, and perception [13,14]. In particular, refraining from going out and attending public gatherings can induce psychosocial stress, and the impact may differ according to sociodemographic variables [15,16,17,18] and resilience levels [16,19,20].

Nonetheless, studies have shown that individual awareness about the outbreak of an infectious disease is an essential factor that influences the implementation of preventive rules [14,21,22,23,24]. Individual awareness can be measured by examining one’s knowledge and attitude towards infectious diseases [14,23,25] or applying the health belief model [13,22,24,26,27].

The following two questions arise amid these considerations: Is there a difference between the psychosocial stress of residents living in areas that experienced a sharp rise in the number of confirmed cases compared to individuals in other areas? Do individual attributes (e.g., knowledge of COVID-19, health belief, and resilience) influence individual psychosocial stress regardless of the increasing number of confirmed cases? Therefore, this study attempted to determine the influence of knowledge of COVID-19, health beliefs, and resilience on psychosocial stress among adults in areas that experience a sharp increase in the number of COVID-19 cases, as well as other areas.

## 2. Materials and Methods

### 2.1. Design and Participants

This study employed a cross-sectional survey design. Adults aged 20–65 residing in the Daegu area (Daegu City and Gyeongbuk Province) and non-Daegu area (entire nation except the Daegu area) for more than six months at the time of data collection participated in the study. The minimum number of participants was calculated as 234 using G*Power 3.1.9.6 software conditioned by multiple regression. Additionally, an effect size (d) of 0.15, a significance level (α) of 0.05, and a statistical power (1-β) of 0.95 were determined.

### 2.2. Data Collection

An online survey was conducted between 19 and 25 May 2020, during the period of “social distancing” to prevent the spread of COVID-19. The structured questionnaire was developed by a researcher using Google Forms, and the recruitment was announced by accessing four communities through one of the most popular websites in Korea as of 29 February 2020. A recruitment announcement was issued on the website accompanied by an explanation, including the purpose of the study, privacy protection, and the time required (5 to 10 min). Interested individuals could click “Agree” to proceed to the questionnaire voluntarily.

### 2.3. Measures

Permission to utilize the measurement tools used in this study was obtained from the original author. The measurements were revised by the authors for this study.

#### 2.3.1. Knowledge of COVID-19

The 7th edition of the “Response Guidance for Local Government to Prevent and Control the Spread of COVID-19”, a tool developed by Park et al. to measure knowledge and attitude towards Middle East respiratory syndrome (MERS), was modified into 13 items to evaluate participants’ knowledge of COVID-19 [28]. One nursing professor and one emergency nurse specialist reviewed the items. The correct answer for each item was assigned one point. Therefore, a higher score indicated greater knowledge of COVID-19. These questions inquired about the causes and symptoms, transmission methods, prevention, and therapy related to COVID-19. The reliability of the items was (KR-21) 0.65.

#### 2.3.2. Health Belief

Based on Erkin and Özsoy’s [21] “Health Belief Model Applied to Influenza (HBMAI)” and Shin’s [22] “Health Belief about Emerging Infectious Disease (HBEID)”, the researchers modified them to be appropriate to the COVID-19 situation. The tool is composed of 29 items with 5 subfactors, including 8 items for perceived susceptibility, 4 for perceived seriousness, 6 for perceived benefits, 8 for perceived barriers, and 3 for cue to action. Each item was measured on a 5-point scale. A higher score indicated more positive health beliefs regarding infectious diseases. The items’ reliabilities (Cronbach’s α) were 0.91, 0.77, and 0.77 in Erkin and Özsoy’s [21], Shin’s [22], and this study, respectively.

#### 2.3.3. Resilience

For resilience, this study employed the Korean version of the Resilience Scale (K-CD-RISC) which was developed from the Connor Davidson Resilience Scale (CD-RISC) [29] and validated by Baek et al. [30]. The tool comprises 25 items with the following 5 subfactors: 9 items for hardness, 8 for persistence, 4 for optimism, 2 for support, and 2 for spirituality. Each item was measured on a 5-point scale (0–4). The score ranged from 0 to 100 and a higher score indicated a higher degree of resilience. At the time of development, the tool’s reliabilities (Cronbach’s α) were 0.89 in [29], 0.91 in the study of Baek et al. [30], and 0.91 in this study.

#### 2.3.4. Psychosocial Stress

To measure psychosocial stress, we used the Psychosocial Well-Being Index Short Form (PWI-SF), based on Goldberg’s General Health Questionnaire (GHQ), modified for the Korean setting by Chang [31]. The tool comprises 18 items measured on a 4-point scale (0–3). A higher score indicated higher stress levels. Participants who scored less than 8 were classified as the “healthy group”, between 9 and 26 as the “potential stress group”, and 27 or more as the “high-risk stress group” [31]. The items’ reliabilities (Cronbach’s α) were 0.90 in Chang’s [31] study and 0.91 in this study.

### 2.4. Ethical Considerations

This study was approved by the Research Ethics Committee of Chung-Ang University (1041078-202002-HR-043-01). The researchers explained the study’s aims, participation methods, careful management of the participants’ data and personal information, and the possibility of participant withdrawal. As soon as the survey period concluded, the URL of the survey was deleted, a digital gift card was sent to each participant, and their personal information was deleted immediately thereafter.

### 2.5. Analysis

For statistical analysis, IBM SPSS Statistics for Windows ver. 25.0 (IBM Corp., Armonk, NY, USA) was used. Independent *t*-test and one-way ANOVA were applied to verify the differences between the variables for each group, Pearson’s correlation coefficients were used to confirm correlations between variables, and multiple regression analysis was used to examine the factors influencing psychosocial stress. We analyzed a total of 338 responses, and there were no missing answers.

## 3. Results

### 3.1. Sociodemographic Characteristics

Participants from the Daegu area and non-Daegu area accounted for 45.0% and 55.0% of the total participants, respectively. Females accounted for 62.4% of the sample and the average age was 39.88 ± 11.48 years (the majority of participants were in their 30s; 39.6%). Most of the participants were married (71.0%), did not have a religious affiliation (64.8%), had graduated from college or higher (60.4%), and were employed (66.3%). Most participants had more than two family members (92.6%) and a monthly household income of more than or equal to five million won (42.0%). Lastly, most of the participants reported that they were healthy (86.4%), and 60.6% received COVID-19-related information from mass media (e.g., TV or newspapers).

### 3.2. Knowledge of COVID-19, Health Belief, Resilience, and Psychosocial Stress

Regarding variables, there were no differences between the participants’ scores from the Daegu area (with concentrated confirmed COVID-19 cases) and the non-Daegu area except for health belief (Table 1).

The average score for knowledge of COVID-19 was 9.40 out of 13, and average correctness was 72%. Among the items, those that demonstrated the highest correctness rate (99.7%) were related to COVID-19 symptoms that varied from mild to severe (fever, sore throat, shortness of breath, pneumonia, etc.). Meanwhile, correctness rate was the lowest in the case of self-quarantine guidelines, such as using a face mask and distancing (10.7%).

The average score for health belief was 3.45 out of 5. There was a significant difference between the Daegu area and the non-Daegu area (*t* = 2.34, *p* = 0.02). Among the subfactors, perceived seriousness (4.31 ± 0.60) was the highest and barriers (2.72 ± 0.63) scored the lowest.

The average score for level of resilience was 63.42 out of 100. When the subfactors were standardized on a four-point scale, support (2.88 ± 0.61) received the highest score and spirituality (2.28 ± 0.75) received the lowest average score.

The average score for psychosocial stress was 33.45 out of 54. The high-risk stress group accounted for 82.0% of participants.

### 3.3. Differences in Knowledge of COVID-19, Health Beliefs, Resilience, and Psychosocial Stress by Sociodemographic Characteristics

The differences between scores for each variable according to sociodemographic characteristics were analyzed using *t*-tests and ANOVA, and Scheffe’s procedure was used for the post-hoc test (Table 2). A significant difference was observed for knowledge of COVID-19 according to age; however, this difference was insignificant in the post-hoc test (*F* = 2.98, *p* = 0.019). Further, those with an education level of graduate school or higher (*F* = 7.07, *p* = 0.001) and those without any religious affiliation (*t* = −2.23, *p* = 0.026) had significantly higher levels of knowledge of COVID-19.

Health belief scores were significantly higher in female participants (*t* = −2.01, *p* = 0.045), participants in the age range of 60 to 65 years (*F* = 3.72, *p* = 0.006), and participants who finished graduate school or higher (*F* = 6.76, *p* = 0.033).

Resilience scores were significantly higher among those with a religious affiliation (*t* = 4.90, *p* = 0.001), those who considered themselves healthy (*t* = 2.40, *p* = 0.020), and those with a monthly household income of less than three million won (*F* = 13.53, *p* < 0.001).

Psychosocial stress scores were significantly higher among participants without a religious affiliation (*t* = 3.09, *p* = 0.002), those who considered themselves unhealthy (*t* = −5.73, *p* = 0.001), and those with a higher monthly household income (*F* = 20.92, *p* < 0.001).

### 3.4. Correlations between Variables

Pearson’s correlation coefficients were applied to identify relationships between variables (Table 3). There was a significant positive relationship between health belief and resilience (*r* = 0.218, *p* < 0.001). Psychosocial stress did not have a significant correlation with knowledge of COVID-19 and health belief, but did have a significant negative correlation with resilience (*r* = −0.578, *p* < 0.001).

### 3.5. Influencing Factors of Psychosocial Stress

Religion, monthly household income, and subjective health status significantly affected psychosocial stress, and were converted into dummy variables. Resilience was used as an independent variable (Table 4). The variance inflation factor value was assessed for multicollinearity, ranging from 1.04 to 1.32; thus, it did not exceed the reference value of 10. The model used in this study was suitable (*F* = 45.50, *p* < 0.001), and its explanatory power was 39.8%. The results revealed that psychological stress was significantly higher when resilience and subjective health status were low, and when monthly household income was high. In particular, resilience had the most significant effect on the level of psychosocial stress.

## 4. Discussion

This study attempted to identify factors affecting Korean adults’ psychosocial stress due to the COVID-19 pandemic. Except for health belief, there was no significant difference between the scores for knowledge of COVID-19, resilience, and psychosocial stress among individuals from the Daegu area (with rapidly increasing confirmed cases) and the non-Daegu area.

The rate of correct answers for questions related to COVID-19 was 72%, which corresponds with the result of a study that confirmed healthcare professionals’ knowledge, including nursing students, when MERS occurred [25,32]. However, no other study on COVID-19 has used the same tools. Knowledge of the causative agent and mortality rate of COVID-19 was low, and this corresponded with the findings of previous studies [24]. While previous studies have observed that the increase in the number of infectious diseases facilitates the need and opportunity to acquire knowledge [17], this study did not find any significant difference between the knowledge of residents from the Daegu and non-Daegu areas. Further, previous studies have determined that the lack of knowledge about infectious outbreaks increases anxiety and depression, which threatens the psychosocial well-being of individuals [13,22]. However, this study did not find a significant correlation between knowledge and psychosocial stress.

The average score of health belief—a variable linked to individuals’ health behavior—was 3.45. This result was higher than the findings of the MERS outbreak study [22]. Among health belief subfactors, the Daegu area had higher perceived susceptibility and seriousness than the non-Daegu area. The non-Daegu area had higher perceived benefits than the Daegu area. However, this could not explain the lack of difference in psychosocial stress between the two regions, although 91% of confirmed cases were reported from the Daegu area. Moreover, those with a religious affiliation attained higher scores, concurring with prior findings, although this was not significant [13]. Health belief did not significantly correlate with knowledge of COVID-19 and psychosocial stress, but showed a positive correlation with resilience (*p* < 0.001). This finding differed from Ye et al.’s conclusion [24] that preventive behavior is associated with knowledge of COVID-19 and subfactors of health belief.

The average score for resilience level was 63.42, which corresponds with previous studies [10,22,30]. The resilience score of those with a religious affiliation was significantly lower than those without, which may be related to negative sentiments against a religious group, Shincheonji—the epicenter of the outbreak in the Daegu area [6]. Due to the current religious stigma, spiritual leaders should collaborate to practice contact-free worship instead of the pre-COVID-19 collective worship [33]. In our study, the average scores for resilience were high among participants with high subjective health status and low monthly household income, which corroborates with prior studies [12,20]. Moreover, resilience showed a negative correlation with psychosocial stress (*p* < 0.001), supporting previous studies that have highlighted the importance of increasing resilience to reduce psychosocial problems [3,8,10,12].

In our findings, the average score of psychosocial stress was 33.45, and the high-risk stress group accounted for 82.0% of participants. This average was higher than the results (22.7 points and 31.3%) of Kim and Cho [18], who used the same measurement tool. Nevertheless, these findings support previous remarks that measures to contain the spread of COVID-19, such as quarantine and lockdowns, may negatively affect psychosocial well-being and act as stressors [9,17]. When considering the severity classification of psychosocial stress among the participants, the high-risk stress group ratio in the non-Daegu area was higher than that of the Daegu area. This seemed to be influenced by unverified information and uncertainty as a result of being distant from the epicenter [34].

Psychosocial stress levels were low among those with a religious affiliation. This supports Milstein’s observation [35] that a religious affiliation is a positive factor for relieving disaster-related stress. Moreover, psychosocial stress was high among those with low subjective health, which may be explained by the fact that chronic patients may suffer from anxiety about overcrowded hospitals or city lockdowns [2,19]. This corresponds with a claim that the lockdown policy may prohibit patients with chronic diseases from visiting hospitals [2], which may increase psychological stress by restricting them from accessing regular checkups or rehabilitation [11].

Further, psychosocial stress was high when participants had a monthly household income of 5 million won or more. As of 2019, the average monthly household income was 3.09 million won in Korea [36]; however, it is difficult to ascertain whether high-income jobs correlate with higher social and psychological stress. Kim and Kang’s study [37] claimed that different classes among Korean workers showed differences in quality of life and psychological health. Among the quarantine policies, enforcing lockdown or shutdown has highly affected small-business workers and employees who rely on public transportation to commute [15,16,38]. Thus, the government’s financial support for individuals should be prudent and effective to facilitate improvements to their psychosocial stress levels. The population of the Daegu area is 1.06 million in Daegu city, and about 1.26 million in rural areas [36], while the non-Daegu areas are mainly residents in the metropolitan area. Due to shutdown rather than the number of confirmed cases, people living in a city with better income experience more barriers to earning money. In the future, it is necessary to determine the level of psychosocial stress by occupational group during a period of strict quarantine and lockdown, implemented in an effort curb COVID-19 [38].

To identify influencing factors, the results of multiple regression revealed that subjective health status, monthly household income, and individual resilience were influencing factors of psychosocial stress (*p* < 0.001), with an explanatory power of 39.8%. The results of this study indicate that those with higher monthly income can experience a more serious negative impact on their resilience and psychosocial wellbeing due to lockdown and shutdown during the COVID-19 pandemic.

## 5. Conclusions

Subjective health, monthly household income, and individual resilience were the influencing factors of psychosocial stress among Koreans during the COVID-19 pandemic. Participants with low subjective health and high monthly household income demonstrated higher levels of psychosocial stress, and those with high resilience demonstrated lower levels of psychosocial stress. Therefore, it is necessary to develop an intervention strategy that promotes community and individual resilience to help individuals cope with the spread of infectious diseases.

However, this study had some limitations. Since the data for this study was collected through an online survey, its generalizability is limited. Nevertheless, this method was suitable because it was economical, simple, and facilitated the remote collection of data. Moreover, we did not conduct longitudinal measurements or explore other potential influencing factors of psychosocial stress. Therefore, further research should be conducted on the effects of the COVID-19 pandemic, especially prolonged quarantine and lockdown, on the psychosocial stress of individuals and communities across different periods. Additionally, comparative studies should be conducted to examine a wide array of potential influencing factors. The results of such studies will facilitate the development of novel interventions and strategies to improve personal and community resilience.

## Figures and Tables

**Figure 1 ijerph-18-06153-f001:**
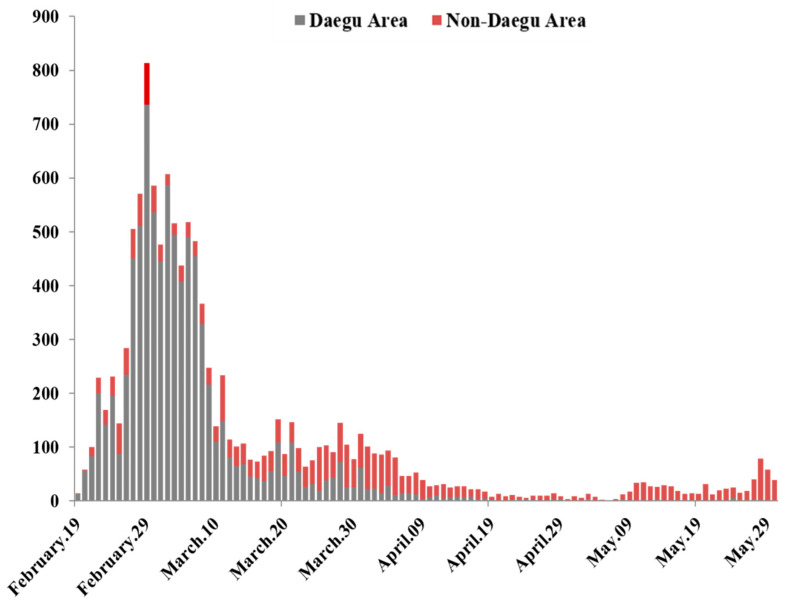
Regional distribution of confirmed COVID-19 cases.

**Table 1 ijerph-18-06153-t001:** Participants’ mean scores for knowledge of COVID-19, health belief, resilience, and psychosocial stress (N = 338).

Variables	Total	Daegu (*n* = 152)	Non-Daegu (*n* = 186)	*t*(*p*)
		Mean ± SD		
**Knowledge of COVID-19**	9.40 ± 1.19	9.32 ± 1.15	9.46 ± 1.23	−1.03 (0.302)
Health belief	3.45 ± 0.34	3.50 ± 0.37	3.41 ± 0.31	2.34 (0.020)
Susceptibility	3.66 ± 0.64	3.91 ± 0.71	3.44 ± 0.48	6.97 (<0.001)
Seriousness	4.31 ± 0.60	4.39 ± 0.56	4.23 ± 0.61	2.38 (0.018)
Benefits	3.85 ± 0.66	3.77 ± 0.76	3.92 ± 0.57	−2.12 (0.034)
Barriers	2.72 ± 0.63	2.66 ± 0.57	2.77 ± 0.64	−1.69 (0.093)
Cue to Actions	2.87 ± 0.84	2.88 ± 0.92	2.87 ± 0.77	0.05 (0.962)
Resilience	63.42 ± 11.23	62.43 ± 11.51	64.23 ± 10.96	−1.47 (0.143)
Hardness	2.41 ± 0.53	2.38 ± 0.52	2.44 ± 0.53	−1.01 (0.312)
Persistence	2.63 ± 0.49	2.56 ± 0.49	2.69 ± 0.48	−2.40 (0.017)
Optimism	2.58 ± 0.60	2.60 ± 0.62	2.57 ± 0.58	0.40 (0.689)
Support	2.88 ± 0.61	2.79 ± 0.67	2.96 ± 0.54	−2.57 (0.011)
Spirituality	2.28 ± 0.75	2.28 ± 0.76	2.29 ± 0.74	−0.07 (0.948)
Psychosocial stress	33.45 ± 8.23	33.03 ± 9.44	33.79 ± 7.10	−0.83 (0.410)
Severity	*n* (%)	*n* (%)	*n* (%)	
Healthy group (≤8)	4 (1.2)	3 (2.0)	1 (0.5)	
Potential stress group (9–26)	57 (16.9)	32 (21.0)	25 (13.5)	
High-risk stress group (≥27)	277 (82.0)	117 (77.0)	160 (86.0)	

**Table 2 ijerph-18-06153-t002:** Differences among participants’ scores for knowledge of COVID-19, health belief, resilience, and psychosocial stress by sociodemographic characteristics.

Characteristic	Category	*n* (%)	Knowledge of COVID-19		Health Belief		Resilience		Psychosocial Stress	
			M ± SD	t/F*(p)* *Sheffé*	M ± SD	t/F*(p)* *Sheffé*	M ± SD	t/F*(p)* *Sheffé*	M ± SD	t/F(*p)* *Sheffé*
Sex	Male	127 (37.6)	9.32 ± 1.22	−1.48 (0.141)	3.42 ± 0.32	−2.01 (0.045)	63.03 ± 11.67	−0.82 (0.413)	33.95 ± 7.97	0.88 (0.381)
	Female	211 (62.4)	9.52 ± 1.13		3.50 ± 0.37		64.06 ± 10.45		33.14 ± 8.39	
Age	20–29 ^a^	65 (19.2)	9.15 ± 1.29	2.98 (0.019) *n.s*	3.39 ± 0.32	3.72 (0.006) *a,b,c* < *e*	62.25 ± 14.44	0.30 (0.879)	31.85 ± 9.21	1.69 (0.152)
	30–39 ^b^	134 (39.6)	9.63 ± 1.21		3.45 ± 0.33		63.59 ± 10.46		33.17 ± 7.44	
	40–49 ^c^	61 (18.1)	9.33 ± 1.21		3.40 ± 0.30		63.51 ± 11.62		33.66 ± 8.70	
	50–59 ^d^	47 (13.9)	9.40 ± 1.17		3.45 ± 0.33		63.53 ± 8.49		33.83 ± 8.24	
	60–65 ^e^	31 (9.2)	9.00 ± 0.58		3.66 ± 0.47		64.77 ± 9.99		33.97 ± 8.04	
Marital status	Unmarried	98 (29.0)	9.32 ± 1.22	−0.79 (0.438)	3.46 ± 0.36	1.16 (0.245)	62.73 ± 11.17	0.64 (0.525)	32.33 ± 8.47	−1.60 (0.110)
	Married	240 (71.0)	9.43 ± 1.18		3.41 ± 0.31		63.70 ± 9.26		33.90 ± 8.11	
Education level	High school ^a^	111 (32.8)	9.14 ± 0.96	7.07 (0.001) *a,b* < *c*	3.54 ± 0.37	6.76 (0.001) *b,c* < *a*	63.37 ± 12.09	0.31 (0.737)	36.43 ± 0.71	1.68 (0.188)
	College/university ^b^	204 (60.4)	9.46 ± 1.26		3.41 ± 0.33		63.25 ± 10.93		33.33 ± 7.43	
	Graduate school ^c^	23 (6.8)	10.09 ± 1.28		3.32 ± 0.29		65.17 ± 9.63		33.04 ± 9.73	
Religious affiliation	Yes	119 (35.2)	9.21 ± 1.06	−2.23 (0.026)	3.47 ± 0.36	0.73 (0.465)	67.34 ± 10.61	4.90 (0.001)	32.44 ± 8.31	−3.09 (0.002)
	No	219 (64.8)	9.50 ± 1.25		3.44 ± 0.34		61.28 ± 10.10		35.30 ± 7.79	
Number of family members	1	25 (7.4)	9.39 ± 1.19	−0.54 (0.591)	3.68 ± 0.40	3.52 (0.055)	64.84 ± 8.65	0.66 (0.511)	31.88 ± 6.43	−1.24 (0.226)
	≥2	313 (92.6)	9.52 ± 1.26		3.43 ± 0.33		63.30 ± 11.41		33.57 ± 8.36	
Occupation	Yes	224 (66.3)	9.54 ± 1.04	1.63 (0.105)	3.42 ± 0.32	−1.74 (0.083)	62.95 ± 11.78	−1.13 (0.261)	33.46 ± 8.88	.044 (0.965)
	No	114 (33.7)	9.33 ± 1.26		3.49 ± 0.39		64.33 ± 10.04		33.42 ± 6.83	
Monthly household income (won/month)	<3 million ^a^	84 (24.9)	9.20 ± 1.23	1.50 (0.225)	3.46 ± 0.34	0.56 (0.573)	66.25 ± 9.20	13.53 (<0.001) *c* < *b* < *a*	29.49 ± 8.51	20.92(<0.001) *a* < *b* < *c*
	3–5 million ^b^	112 (33.1)	9.45 ± 1.23		3.46 ± 0.34		63.51 ± 12.26		32.78 ± 8.37	
	>5 million ^c^	142 (42.0)	9.47 ± 1.13		3.41 ± 0.35		58.50 ± 11.43		36.32 ± 6.80	
Subjective health status	Healthy	292 (86.4)	9.43 ± 1.15	1.05 (0.296)	3.48 ± 0.37	0.62 (0.535)	64.04 ± 10.71	2.40 (0.020)	27.24 ± 7.91	−5.73 (0.001)
	Unhealthy	46 (13.6)	9.20 ± 1.42		3.44 ± 0.34		59.48 ± 12.13		34.42 ± 7.86	
Channel of COVID-19 information	Mass media	205 (60.6)	9.70 ± 0.71	0.36 (0.698)	3.34 ± 0.24	0.62 (0.538)	62.13 ± 8.68	0.10 (0.908)	33.50 ± 7.53	0.62 (0.539)
	Internet/SNS	125 (37.0)	9.39 ± 1.20		3.47 ± 0.34		63.68 ± 12.51		33.57 ± 9.21	
	Acquaintances/hospitals	8 (2.4)	9.39 ± 1.20		3.44 ± 0.35		63.31 ± 10.50		30.25 ± 8.23	

**Table 3 ijerph-18-06153-t003:** Correlations between participants’ scores for knowledge of COVID-19, health belief, resilience, and psychosocial stress.

Variable	1	2	3	4
1. Knowledge of COVID-19	1			
2. Health belief	−0.057 (0.294)	1		
3. Resilience	0.034 (0.537)	0.218 (<0.001)	1	
4. Psychosocial stress	−0.059 (0.227)	−0.103 (0.060)	−0.578 (<0.001)	1

**Table 4 ijerph-18-06153-t004:** Influencing factors of psychosocial stress.

Variables	*B*	*SE*	*β*	*t*	*p*
(Constant)	11.95	2.23		5.36	<0.001
Resilience	−0.37	0.03	−0.51	−11.16	<0.001
Religion *	−0.05	0.76	−0.01	0.07	0.943
Subjective health status **	−4.77	1.04	−0.20	−4.59	<0.001
Monthly household income ***					
<3 million	−3.22	0.92	−0.17	−3.48	0.001
3~5 million	−2.01	0.82	−0.12	−2.45	0.015
*R*^2^ = 0.368, *Adj R*^2^ = 0.398, *F* = 45.50, *p* < 0.001, Durbin–Watson = 1.98					

* Dummy code (reference = no religion). ** Dummy code (reference = unhealthy). *** Dummy code (reference = ≥5 million).

## Data Availability

Not applicable.

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
