# Peer review of "Influencing Factors of Psychosocial Stress among Korean Adults during the COVID-19 Outbreak"

_ijerph, 2021, doi:10.3390/ijerph18116153_

Round 1

Reviewer 1 Report

Thank you for the opportunity to review the manuscript: Influencing Factors of Psychosocial Stress Among Korean Adults During the COVID-19 Outbreak. This topic is timely considering the influence of COVID-19 on lifestyle factors. The article is well thought out regarding the measures, the analysis and the discussion but could be improved through the following comments.

Major points

  1. The readability of the manuscript is problematic. There are minor grammatical errors that hurt the flow of the document such as “To measure knowledge of COVID-19, a tool measuring…line 83. has multiple measure words that is cumbersome and redundant. Others include have vs had and some sentences would be better broken into two sentences. I would suggest an independent party review the wording.

  1. What counted as consent? Did they click a link or an “Agree” button or did they write in their initials or name first? What was the time commitment to answer all the questions?

  1. Table 1 needs more information so it can stand alone such as a score out of total so the reader can get a better sense of the ranges. More importantly, the scope of the work mentioned in line 51 does not fully match the results. I expected to have Table 1 expanded to the Daegue and non-Daegue areas with reported p-values to denote differences in total scale and subscale measures.

  1. The discussion does not adequately discuss the findings between the Daegue and non-daegue areas except stating there were no differences. Were the populations similar with respect to the demographics. This information is not given. Why would there be such similarities.

  1. The discussion attempts to link future intervention tactics with the examined variables. The implications of the results are not very developed with many of the variables being non-modifiable. Please expand on this topic.

Minor points

Abstract:

Line 8-9, The text size is mixed in the abstract.

Line 11, The term outbreak should be replaced with pandemic

The correlations will require statistical values placed in the abstract

Measures:

Line 84, MERS needs to be written out at first mention and then abbreviated afterward.

Results:

Line 152, I don’t understand how an average can have a range. Try rewording.

Line 179 The table and the knowledge due to occupation don’t match the way I am reading it.

Line 195 Awkward sentence.

Conclusion:

Restating the results in the conclusion is redundant

Author Response

Thank you for your valuable comments. Please see attached. 

Reviewer 2 Report

Surprising that increased income would present as a risk factor for psychosocial stress.  Would like to see further exploration of this in discussion with comparison to other studies with parallel findings.

Author Response

Thank you. Please see attached file. 

Reviewer 3 Report

The authors studied influencing factors of psychosocial stress among Korean adults during the COVID-19 outbreak. The study’s topic is relevant to International Journal of Environmental Research and Public Health. The study was well planned and executed. Data analysis was appropriate and discussion and conclusions were well written. Nevertheless, there were some issues (numerical errors) in the article that the authors should address.

  1. Line 12, Abstract: “…334 adults …” should be “…338 adults…” (Please see line 137, Table 1, Table 2, etc.).
  2. Line 156, Section 3.2: The authors reported that “The average score for knowledge of COVID-19 ranged from 9.40-13”. Please check this statement. It should be noted the average score should be a number (but not a range).
  3. Line 156, Section 3.2: “…using a face mask and distancing (10.7%%)” Please delete the second ‘%’. Additionally, please check whether there was a percentage for using a face mask.
  4. Line 161, Section 3.2: The authors stated that “The average score for level of resilience was 61.28+/-10.71 (95% CI: 60.14, 62.43)…” These values were incorrect because the authors reported that the resilience level was 63.4 later in the article (line 248). I looked at the components of resilience in Table 1. I calculated that the average score should be 63.41 (=2.41x9+2.63x8+2.59x4+2.88x2+2.28x3). Please recalculate CI values.
  5. Line 179, Section 3.3: The statement “…no occupation (t = 2.65, p = 0.009)” should be “…having occupation (t = 2.65, p =0.009)”. Please look at Table 2 more carefully. Table 2 shows that people having occupation (for ‘Yes’) had a higher level of knowledge of COVID-19 (9.52 +/- 1.13).
  6. Line 187, Section 3.3: The statement “…less than 3 million won…” should be “…more than 5 million won…” Please look at Table 2 more carefully. Table 2 shows that people having more than 5 million won of monthly household income had a higher score of resilience at 66.25+/- 12.26.
  7. Table 2 contained many errors such as
  • For knowledge of COVID-19 (Column 4), the average score should be 9.4. However, the values for the number of family (1 and 2) were 9.52… and 9.46…, respectively. Please note that it was impossible to have both numbers to be greater than 9.4 that was the average score. Please check your data file.
  • For health belief (Column 6), the average score should be 3.58 (see line 238). However, the values for ‘male’ and ‘female’ were 3.56 … and 3.47 …, respectively. Please note that it was impossible to have both numbers to be less than 3.58. Additionally, the numbers for ‘marital status (unmarried or married)’, ‘region (non-Daegue area or Daegue area)’, ‘religious affiliation (yes or no)’, ‘occupation (yes or no)’, ‘monthly household income (<3 million, 3-5 million, or >= 5 million)’, ‘subjective health status (healthy or unhealthy)’, and ‘channel of COVID-19 information (mass media, internat/sns, or acquaintances/hospitals)’ were all less than 3.58. Specifically, the last category – ‘channel of COVID-19 information’ had values of 2.55…, 2.53…, and 2.45 … which were clearly wrong.
  • For psychosocial stress (Column 10), the average score should be 33.45 (see Table 1 or line 259). However, the values for the number of family (1 or >=2) were 30.52… and 31.88… which should be incorrect. Please check your data.
  1. Line 196, Section 3.4: ‘Table 5’ should be ‘Table 3’.
  2. There were some inconsistencies in References. Please only capitalize the first word of the title in References 10, 12, 32, and 35.
  3. After revising Tables 1 and 2, I suggest the authors to check the results shown in Table 3 (correlations between constructs) and Table 4 (regression analysis).

Author Response

Please see attached file. Thank you for your valuable comments. 

Round 2

Reviewer 1 Report

Thank you for the opportunity to rereview this manuscript. the authors have done a good job in addressing many of the issues first noted. Besides moving the limitations prior to the conclusions, I have no other methodological concerns regarding the rigor of the study.